# Influence of Gamification and Cooperative Work in Peer, Mixed and Interdisciplinary Teams on Emotional Intelligence, Learning Strategies and Life Goals That Motivate University Students to Study

**DOI:** 10.3390/ijerph20010547

**Published:** 2022-12-29

**Authors:** Celia Redondo-Rodríguez, José Alberto Becerra-Mejías, Guadalupe Gil-Fernández, Francisco José Rodríguez-Velasco

**Affiliations:** 1Department of Psychology and Anthropology, Teacher Training College, University of Extremadura, Av. de la Universidad, s/n, 10071 Cáceres, Spain; 2Department of Nursing, University Center of Plasencia, University of Extremadura, C. Virgen del Puerto No. 2, 10600 Plasencia, Spain; 3Department of Nursing, Faculty of Medicine and Health Sciences, University of Extremadura, Av. Elvas, s/n, 06006 Badajoz, Spain

**Keywords:** gamification, cooperative, learning strategies, life goals, emotional intelligence

## Abstract

It is necessary to motivate university students to reduce the dropout rate in Spain, and to look for strategies that help university students acquire professional competencies; this is where gamification can be useful. The purpose of the study was to evaluate the influence of a learning methodology based on gamification and cooperative work in peers, and in mixed and interdisciplinary teams on the emotional intelligence, learning strategies, and life goals that motivate university students to learn. The sample consisted of 102 students who took a subject with a gamification-based methodology, through the Mechanics-Dynamics-Aesthetics model, in a cooperative way. The Self-Perception Scale of Personal Academic Motivation and the Trait Meta Mood Scale 24 were used. The results of the study showed an increase in learning strategies and life goals that motivated university students to study, as well as increases in emotional clarity and significantly in emotional repair. It was concluded that gamification is a positive tool for its ability to increase emotional intelligence, life goals, and learning strategies in university students’ motivation to learn.

## 1. Introduction

New innovative approaches and techniques are continually being sought to be applied in education at all levels, including university, to improve both teaching and learning [1]. Although gamification-based educational interventions are becoming increasingly popular [2], there is still no clear understanding of when gamification can be an appropriate tool for instruction and learning [3]. Gamification is understood as the application of game-like elements to real-world processes, issues, and problems [1].

While it is true that gamification has a lot of potential in education [4], the results obtained on the influence of gamification are still not definitive or conclusive [5]. Furthermore, most of the studies conducted on gamification have focused on virtual interventions, leaving aside gamification-based educational interventions carried out in face-to-face classes [5]. Therefore, gamification must be evaluated in a very rigorous way to check for possible benefits that it may cause, and to what extent that it could do so [1].

It is necessary to look for strategies that help university students to acquire the competencies indicated in the academic curriculum, as well as their professional competencies; this is where gamification can be useful for university students [6], as it has been shown that it can be effective for the future work performance of these students [7]. Very recent research affirms that gamification techniques that are addressed through cooperative work techniques are effective in pro-moving and developing teamwork [8] in interdisciplinary environments [9], that is, those in which members of different disciplines come together and agree on the application of terms, approaches, or a methodology, thereby integrating their work and procedure [10]. Cooperative learning strategies seem to positively influence the development of students’ social skills competences [11,12,13], and these can also be enhanced through gamification [14], thus enabling students to train for their professional future through acquiring competences related to teamwork [13].

Emotional intelligence (EI) is considered to be a further professional competence that includes the ability to accurately perceive, appraise, and express emotion; the ability to access and/or generate feelings when facilitating thoughts; the ability to understand emotion and emotional knowledge; and the ability to regulate emotions to promote emotional and intellectual growth [15]. The labor context demands from graduating university students are for them to not only have knowledge, but also to be carriers of socioemotional skills that allow them to successfully face future labor problems. EI in the university environment facilitates both the training process and the professional success of future graduates [16]. For this reason, this study considered the fact that a technique such as gamification, combined with cooperative work can develop students’ EI, which influences improvements in learning outcomes [17,18] and reduces the tendency to drop out of university studies [19]. This phenomenon is of great interest due to the need to mitigate the economic, personal, and social effects it generates [20].

In the Spanish university education system there is a generalized lack of motivation among students [21]; this can be seen in the fact that dropout rates are around 30% [20]. However, according to the authors’ knowledge and understanding, gamified cooperative activities can improve motivation for tasks [8,22]. Moreover, cooperative learning strategies can lead learners to more adaptive motivational patterns [12], as is the case with gamification, since, according to the results of several studies, it increases motivation [14], reducing the number of students who drop out of their studies [23]. Less information is found in the domains related to the motivation to learn, such as life goals and learning strategies, which were the subjects of study in this research. Boza and Méndez defined life goals, linked to the motivation to learn, as those that have to do with achieving a better future, life security, professional competence, economic and personal success, and satisfaction with knowledge [24]. On the other hand, learning strategies can be understood to be the organized, conscious, and intentional set of what the learner does to achieve his or her goals, in a given social context [25].

Despite the existence of multiple research on gamification in the international field, there are few scientific studies in our country regarding the influence of educational methodologies based on gamification [26]. The main objective of the present study was to verify the influence of a learning methodology based on gamification and cooperative work in peers, and in mixed and interdisciplinary teams on university students, and specifically to address the following:To assess the influence of gamification and cooperative work on the EI of university students;To assess the influence of gamification and cooperative work on the life goals and learning strategies that motivate university students to study.

## 2. Materials and Methods

### 2.1. Study Design, Setting and Participants

A descriptive observational study was carried out by the EDUCAEN Teaching Innovation Group and Biopsychosocial Research Group (GIBIPSO) of the University of Extremadura during the second semester of the 2020/2021 academic year. The sample consisted of 102 students (78 women and 24 men); 50 students belonged to the Degree in Primary Education (38 women and 12 men), and 52 students belonged to the Degree in Psychology (40 women and 12 men). The ages of all the subjects in the sample were between 18 and 29 years, with the mean age being 18.70 ± 1.70; the mean age of the female subjects was 18.45 ± 1.02, and the mean age of the male subjects was 19.50 ± 2.90.

All subjects who participated in the study met all inclusion and exclusion criteria, and were recruited from the first class of the subject, Cognitive and linguistic development in the Psychology degree, as well as the subject, Psychoeducational attention to diversity and school coexistence in the Primary Education degree; these were classes in which the subjects voluntarily answered the questionnaire. In both degrees, an educational innovation program was implemented through gamification and cooperative work in peer, mixed, and interdisciplinary teams. On the last day of class, the university students voluntarily returned to answer the questionnaire.

The whole process was carried out following the Declaration of Helsinki, and ethical approval was obtained from the Bioethics and Biosafety Commission of the Vice-rectorate for Research and Transfer of the University of Extremadura (Register no. 196/2020-approved on 14 December 2012). The method of choice of participants was non-probabilistic purposive sampling; prior to data collection, students were informed of the nature of the study, and their anonymity was assured. All participants provided oral and written informed consent to participate in the study and for subsequent publication of their anonymous responses.

### 2.2. Instruments

A quantitative research design was carried out using the Personal Academic Motivation Self-Perception Scale [24], which has a Cronbach’s alpha of 0.855 for life goals and 0.831 for learning strategies, and the Trait MetaMood Scale 24 (TMMS-24) adapted to Spanish by Fernández-Berrocal [27]. According to the authors, Cronbach’s alpha for the emotional attention dimension is 0.90, for emotional clarity is 0.90, and for emotional repair is 0.86. Both questionnaires were answered by students in an online format, via estudioenfermeria.com.

The evolution of EI of university students, was measured in terms of the mean difference between the beginning and end of the course in the TMMS-24 scores, consisting of a five-point Likert-type scale (from 1 = Not at all agree to 5 = Strongly agree).

The choice of the Self-Perceived Personal Academic Motivation Scale was made with the purpose of finding out the evolution of life goals (cognitive, self-affirming, task, affective, and social) and learning strategies (prior knowledge, global readings, adequate sources, synthesis, various resources, personal readings, relevant contents, and annotations) in the learning motivation of university students, assessed on a Likert-type scale ranging from 1 (do not agree at all) to 7 (strongly agree) [24]. Learning strategies are defined as the operations performed by the thinking process when faced with the task of learning. Strategies have an intentional character and therefore, involve a plan of action [28].

Life goals are understood as a cognitive representation of what an individual is trying to achieve in a given situation [29]. For this reason, they represent the consequences to be achieved or avoided, and direct the other components of the person in trying to bring about those consequences, or prevent them from occurring [30]. Specifically, the life goals that have been assessed in university students are as follows: affective life goals consisting of studying to feel active, to avoid boredom, to calm down, and to avoid stress, to be happier, to feel satisfied, to feel healthy, strong and energetic; cognitive life goals based on studying to know, to understand the subject in depth, to develop intellectual creativity, and to improve self-confidence; self-assertive life goals consisting of studying to feel unique, different from others, to be freer in opinions and decisions, to be better than others, to be valued by others, and to integrate into society; task goals consisting of studying for a better future, a secure life, success in life, and earning money; and finally, social life goals consisting of studying to take social responsibility, to promote justice and fairness, to be able to help others, and to be competent in the subject or task.

The TMMS-24 was used to measure the EI of university students participating in the cooperative gamification experience in the teaching–learning process. The scale is composed of 24 items, which are grouped into the following dimensions: emotional attention, i.e., the ability to feel and express feelings appropriately; clarity of feelings, understood as the correct understanding of emotional states; and finally, emotional repair, i.e., the ability to regulate emotional states [31,32].

The online modality favored anonymity as a form of expression, which allowed students to express themselves freely [33].

### 2.3. Procedure

A gamified and cooperative methodology, “*Winter is coming*”, was designed and evaluated for the first-year compulsory subjects of Cognitive and linguistic development in the Psychology degree, and Psychoeducational attention to diversity and school coexistence in the Primary Education degree, in the second semester of the 2020/2021 academic year.

In the classes, a methodology was implemented that consisted of taking advantage of the motivational principles of games in the adaptation of the filmic reference “*Game of Thrones*”, in order to promote learning strategies, vital goals in the motivation to learn, and the EI of the students.

The cooperative gamification program was developed in three weekly sessions that lasted one and a half hours for two sessions and one hour for the other session. It was structured within the Mechanics-Dynamics-Aesthetics (MDE) model [34], where the aesthetic was based on the narrative of the Game of Thrones series; the dynamics consisted of challenges and cooperative activities, and the mechanics used resources.

The program contained a narrative that offered a story with roles for each of the students in the subject, and the formation of working groups of 5 to 6 students, called family houses.

The project was presented to the students as a live board game, where the fiction conditioned their reality during this learning experience. To do this, a digital board was used, designed with the Microsoft Paint image editor program. The squares were the territories of Westeros. Each of these squares represented a theme of the subject where the students had to conduct a series of activities and challenges in a cooperative way to reach the final square and achieve the objective of the adventure (Figure 1).

The rules of the game consist of the fact that, depending on the evidence of learning in each work group, students can obtain resources that affect the game based on the activities and challenges they carry out; they have to put into practice the contents and competences worked on in the classroom sessions that correspond to the teaching guide for the subject.

In addition to the cooperative activities related to the content of the subject, students were given challenges that had to be conducted by unifying the homologous family houses of the Psychology degree and the Primary Education degree, in order to form interdisciplinary work groups. One of the challenges consisted of making a challenge via Twitter go viral with the hashtags *#UEXPsicologíaGOT* and *#UEXEdu-PrimariaGOT* for approximately four hours, with the aim of highlighting those characters who deserved to be honored for being an example of some characteristic that identifies a good psychologist or teacher, in order to raise awareness as future professionals about the values, characteristics, and skills to work on to become great professionals.

The rest of the challenges consisted of deciphering riddles, completing word searches, puzzles, and riddles that were interrelated to the content of the subjects of Cognitive and linguistic development and Psychoeducational attention to diversity and school coexistence. Communication between the interdisciplinary groups was via official emails of the university institution.

The resources were exchanged for points in the final grade of the subject. The teaching role during the course consisted of the role of pedagogical accompaniment [35], where, during each session, the doubts of each group of students were guided.

The gamification was designed jointly by the teacher of the subject and one of the main researchers. Before the implementation, the teacher conducted a three-and-a-half-hour training session on gamified learning strategies. The necessary resources for gamification were provided.

### 2.4. Statistical Analysis

The methodology followed for recording the sample data was as follows: firstly, all the variables analyzed were recorded on the study record sheet, and then in the database of the IBM SPSS Statistics 24 statistical application, for the statistical analysis of the results.

The parametric paired-samples *t*-test was performed to compare the means of EI, learning strategies, and vital goals in the motivation to learn of university students, at the beginning of the compulsory subject, in the first year of the degree, and at the end of it. The statistical analysis was conducted with a confidence level of 95% and significant values were considered as *p* < 0.05. The skewness and kurtosis values of all variables were within the acceptable limit of ±2 (ranging between 0.44–1.21 and 0.12–0.90, for skewness and kurtosis, respectively), indicating a normal distribution [36].

## 3. Results

### 3.1. Study Sample

The initial sample consisted of 147 students belonging to the University of Extremadura, where an educational innovation program was implemented through gamification and cooperative work in peer, mixed, and interdisciplinary teams. In the first and last sessions, students volunteered to answer an online survey. This survey was completed by 102 students (69.3% of the participants), 78 females, and 24 males aged 18–29 (18.70 ± 1.70). There was a lower number of men due to the demographic characteristics of the entry profile to the Psychology and Primary Education degrees. In our final intake, 50 students were studying for a bachelor’s degree in Primary Education, and 52 students were studying for a bachelor’s degree in Psychology (Figure 2).

The method used to select the participants was non-probabilistic purposive sampling. The inclusion criteria for participation in the study were as follows: taking the compulsory subject of Cognitive and Linguistic Development in the first year of the Psychology degree or Psychoeducational Attention to Diversity and School Coexistence in the Primary Education degree, during the 2020/2021 academic year; attending more than 50% of the classes; and signing the informed consent form.

After analyzing the surveys completed by university students who took a compulsory subject in the first year of the degree, with a methodology based on cooperative gamification, the following results were identified in EI, learning strategies, and life goals that motivated the study (See Table 1).

Table 1 shows the pre- and post-scores obtained on the three dimensions of EI (emotional attention, emotional clarity, and emotional repair), on the learning strategies and the life goals motivating the sample study.

### 3.2. Evolution of EI

The results show that university students who experienced a gamified and cooperative environment in peer, mixed, and interdisciplinary teams, through the MDE model, increased the mean score in emotional clarity (−1.44 ± 5.07, *p* = 0.005) and emotional repair (−1.37 ± 4.09, *p* = 0.001). Regarding emotional attention, there were no significant changes in the pre-post mean difference (0.17 ± 4.43, *p* = 0.688), with university students maintaining adequate emotional attention. Cronbach’s alpha for all constructs was greater than 0.82.

### 3.3. Evolution of Factors Associated with Motivated Learning among University Students: Learning Strategies and Life Goals

As can be seen from the results, cooperative gamification, through the MDE model, promoted changes in the participants by the end of the course, increasing the use of their learning strategies (0.40 ± 0.93, *p* = 0.000). Statistically significant differences were found between the means of the following sub-variables of the learning strategies construct: global readings pre (4.18 ± 1.67) and global readings post (4.84 ± 1.72); adequate sources pre (4.86 ± 1.30) and adequate source post (5.30 ± 1.32); synthesis pre (4.57 ± 1.66) and synthesis post (5.11 ± 1.53); different resources pre (4.13 ± 1.33) and different resources post (4.54 ± 1.49); personal readings pre (3.44 ± 1.53) and personal readings post (3.80 ± 1.64); and relevant content pre (4.92 ± 1.51) and relevant content post (5.36 ± 1.43) (Table 2).

On the other hand, university students also increased the vital goals in their motivation to learn (0.13 ± 0.66, *p* = 0.040) at the end of the course, compared to the life goals they had at the beginning of the course. Statistically significant differences were found in the mean scores of the sub-variables of the life goals construct, including the following: self-assertiveness pre (3.58 ± 1.32) and self-assertiveness post (3.79 ± 1.49); affective pre (3.13 ± 1.23) and affective post (3.45 ± 1.30) (Table 2).

## 4. Discussion

In recent decades, EI has acquired great relevance due to the fact that its training and development equips students with the necessary competencies to cope with the various situations that arise throughout their academic careers [37]; in addition, those students who have higher levels of EI have a better self-perception concerning their personal growth [38], and manage to avoid the wear and tear caused by the academic environment that surrounds them daily, in particular, with respect to restlessness and anxiety [37]. However, students must also be prepared for the requirements of the labor market. Therefore, in addition to acquiring basic competencies by taking the degree subjects, students need to acquire interdisciplinary competencies that are useful for future job performance [39]. It has been shown that gamification as a pedagogical mediation tool can encourage the development of different skills that are especially needed in work environments [40].

According to these data, the use of gamification in the classroom, as an educational methodology through the MDE model in a cooperative way in teams, promoted changes in two dimensions of EI (emotional clarity and emotional repair) in the members of the peer, mixed, and interdisciplinary teams, confirming the hypothesis. That is, students improved their perceived understanding of their emotional states and their perceived ability to regulate their emotional states correctly. Concerning emotional attention, students maintained the ability to feel and express emotions appropriately, as a very low or very high score in emotional attention shows that too little or too much attention is paid to emotions, indicating problems of different types in the subject [41]. Our results suggest that gamification and cooperative learning in peer, mixed, and interdisciplinary teams promote the acquisition of skills needed for job performance through developing EI [42,43]. These results are not consistent with those obtained in another study in which the findings showed that there was no significant improvement in EI after implementing gamification strategies in the classroom [44]; however, the results are consistent with other studies showing that EI is promoted when game-based teaching strategies [45,46,47] or cooperative learning are implemented in the classroom [12,48].

Thus, advances in education should be based on measures that are reinforced by an interdisciplinary approach that allows educational and work activities to be aligned with each other, creating an environment in which gamification could be used as a tool to improve performance [49]. The evidence consulted shows that the implemented programs are effective when they have a medium-length implementation period, such as the one carried out in this study, while those of short duration are not effective [37].

In our study, we detected that the use of gamification in the classroom, through the MDE model, in a cooperative way in peer, mixed, and interdisciplinary teams, influenced an increase in learning strategies and life goals that motivate university students to study, thus confirming the hypothesis. Specifically, in life goals, the post-intervention increase in the mean scores of the “self-assertive”, and “affective” sub-variables was significant. In addition, in learning strategies, the post-intervention mean scores of the sub-variables “global readings”, “adequate sources”, “synthesis”, “different resources”, “personal readings”, and “relevant content” increased.

Both are dimensions that are associated with motivated learning in university students [24], thus an increase in university students’ life goals and learning strategies will have a positive impact on students’ motivation to study [50,51] contributing to university retention [23]. In the sample of the present study, 100% of the students did not intend to abandon their university studies after completing the course with a methodology that was based on gamification and cooperative work, so the promotion of projects such as this one by the higher education institution can help to increase the permanence of students to continue their university studies [52]. Currently, there is controversy on the effect of the influence of motivation on learning strategies. A recent study by Santoro et al. found no significant effect of motivation on learning strategies [53], while the results obtained by Hongsuchon et al. showed that students who had higher levels of motivation had a greater ability to adopt learning strategies more effectively [54]. The authors of the articles contained in a recent systematic review are along the same lines as the results of the present study, stating that gamified methodologies and systems ensure a better experience for the participant, greater motivation, and greater commitment [55]. This is also emphasized by a recent meta-analysis which indicated that gamification-based educational interventions are effective in promoting student learning and motivation, but also suggested that due to the small number of articles on the subject, more studies were needed to confirm these findings [5].

Despite the results obtained, some limitations of this study are discussed. Firstly, the sample was not probabilistic, and the subjects belonged to the same university. Secondly, it should be noted that the results of the TMMS-24 reflected the self-rated subjects’ perception of their attention to feelings, emotional clarity, and emotion repair. It is, therefore, possible that some subjects overestimated their abilities, while others underestimated them. Thirdly, it should be borne in mind that the current entry profile in the degrees where the study was carried out meant that the sample was made up of many more women than men.

On the other hand, it is worth noting that learning strategies are one of the least-studied study techniques currently, and this is considered one of the main features of the present study. The choice of basing gamification on the filmic reference *“Game of Thrones”*, is because it is characterized by everything that shapes the life of any professional in the field of psychology or education, since students constantly have to agree on decisions, be able to adapt to changing situations, and manage the consequences and emotions that arise from them, in order to be able to execute the challenges posed in a cooperative manner, in the best possible way. These characteristics and situations do not usually occur in university classrooms, but they have an educational value beyond any doubt, and what better learning context can there be than one that enhances, on the one hand, life goals and learning strategies in the motivation to study of university students, and on the other hand, students’ EI?

As a future line of research, it would be interesting to be able to extend this experiment to the university community, in order to contrast data with other universities, and thus improve the model through experience.

## 5. Conclusions

Our results suggest an improvement in EI, life goals, and learning strategies in the learning motivation of university students, when taking a subject with a gamification-based methodology, through the MDE model, in a cooperative way, for 15 weeks. Therefore, we recommend the use of cooperative play strategies in the university educational context as a vehicle to increase students’ academic motivation, which may have an impact on students’ academic performance and, consequently, reduce the number of students who drop out of university studies.

On the other hand, given that the cooperative gamified methodology increased the average scores of useful learning strategies in different areas of knowledge (global reading, selection of appropriate sources, synthesis of contents, use of different resources and complementary materials, personal reading, and the ability to distinguish relevant contents), and also increased the average levels of self-assertion and affective life goals of university students, we suggest that students from different fields of knowledge could benefit from the application of this type of teaching methodology.

In this sense, we believe that gamification and cooperative learning can help university students acquire professional skills that are especially relevant for working cooperatively in interdisciplinary teams, and are of particular interest for their integration into university classrooms in areas of knowledge that are developed in this type of context.

For all these reasons, we believe that the academic and work-related benefits suggested by this type of teaching methodology justify the effort and time required by teachers to implement it in the classroom.

## Figures and Tables

**Figure 1 ijerph-20-00547-f001:**
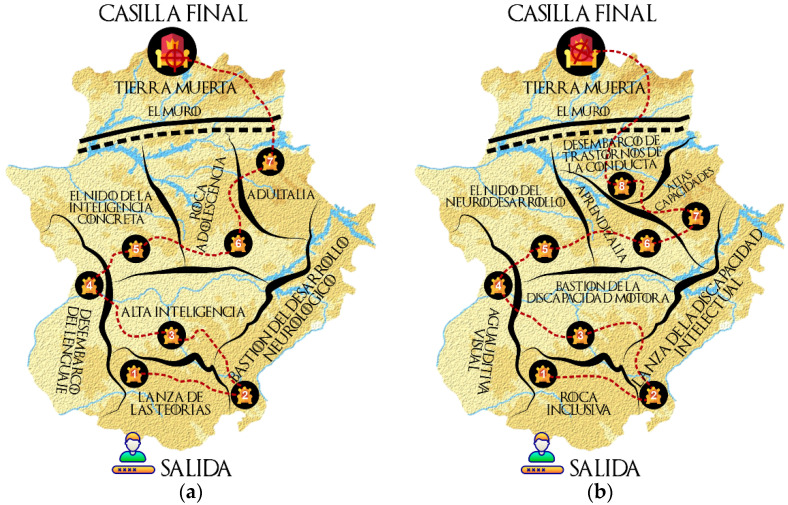
Digital dashboard corresponding to the territories of Westeros of the objectives of the Degree in Psychology (**a**) and the Degree in Primary Education (**b**).

**Figure 2 ijerph-20-00547-f002:**
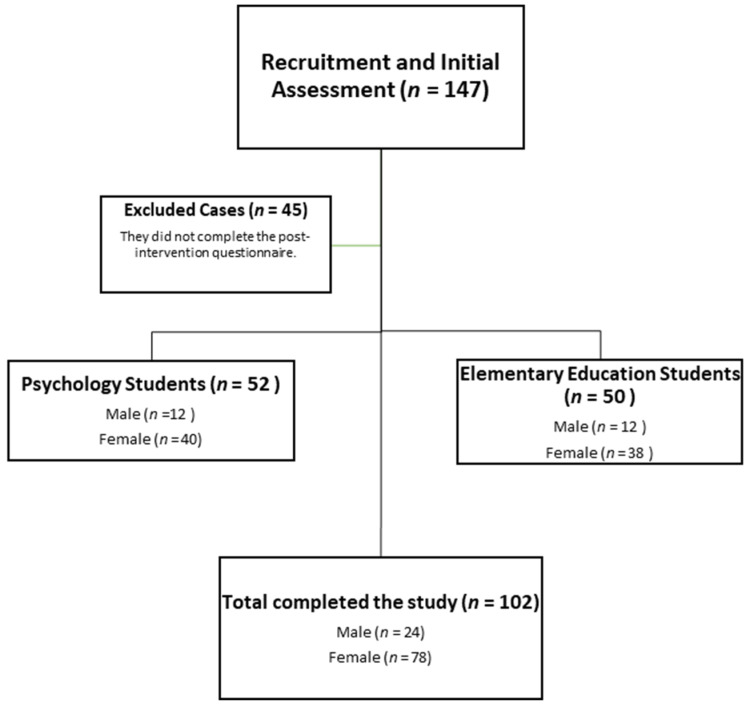
CONSORT flowchart of recruited students.

**Table 1 ijerph-20-00547-t001:** Pre- and post-scores for EI, learning strategies, and life goals motivating study of the study subjects.

	N	M (SD)	Min	Max	t	*p*-Value
**Emotional Intelligence**						
Attention pre	102	28.43 (±5.43)	14.00	38.00	0.40	0.688
Attention post	102	28.25 (±5.56)	15.00	40.00		
Clarity pre	102	24.05 (±5.85)	14.00	40.00	−2.87	0.005
Clarity post	102	25.50 (±6.45)	11.00	40.00		
Repair pre	102	23.91 (±5.95)	12.00	37.00	−3.39	0.001
Repair post	102	25.28 (±5.94)	14.00	40.00		
**Motivated Learning**						
Learning strategies pre	102	4.44 (±1.05)	2.10	7.00	−4.42	0.000
Learning strategies post	102	4.85 (±1.15)	2.60	7.00		
Life goals pre	102	4.52 (±0.88)	2.55	6.36	2.08	0.040
Life goals post	102	4.65 (±1.01)	2.05	6.77		

M: mean; SD: standard deviation; N: total sample; Min: minimum; Max: maximum; t: paired samples *t*-test.

**Table 2 ijerph-20-00547-t002:** Pre- and post-scores for the sub-variables of learning strategies and life goals motivating the study of study subjects.

	N	M (SD)	Min	Max	t	*p*-Value
**Life Goals**						
Cognitive pre	102	4.90 (±1.37)	1.50	7.00	−1.37	0.173
Cognitive post	102	5.06 (±1.47)	1.25	7.00		
Self-assertiveness pre	102	3.58 (±1.32)	1.40	7.00	−2.01	0.047
Self-assertiveness post	102	3.79 (±1.49)	1.20	7.00		
Task pre	102	5.99 (±1.21)	1.50	7.00	−0.80	0.428
Task post	102	5.92 (±1.19)	1.75	7.00		
Affective pre	102	3.13 (±1.23)	1.00	6.20	−3.17	0.002
Affective post	102	3.45 (±1.30)	1.00	6.80		
Social pre	102	5.60 (±1.15)	2.25	7.00	−0.11	0.916
Social post	102	5.61 (±1.22)	2.25	7.00		
**Learning Strategies**						
Prior knowledge pre	102	5.50 (±1.33)	2.00	7.00	−1.82	0.071
Prior knowledge post	102	5.72 (±1.29)	2.00	7.00		
Global readings pre	102	4.18 (±1.67)	1.00	7.00	−3.48	0.001
Global readings post	102	4.84 (±1.72)	1.00	7.00		
Adequate sources pre	102	4.86 (±1.30)	2.00	7.00	−3.77	0.000
Adequate sources post	102	5.30 (±1.32)	2.00	7.00		
Synthesis pre	102	4.57 (±1.66)	1.00	7.00	−3.49	0.001
Synthesis post	102	5.11 (±1.53)	1.00	7.00		
Different resources pre	102	4.13 (±1.33)	1.00	7.00	−2.99	0.004
Different resources post	102	4.54 (±1.49)	1.50	7.00		
Personal readings pre	102	3.44 (±1.53)	1.00	7.00	−2.59	0.011
Personal reading post	102	3.80 (±1.64)	1.00	7.00		
Relevant content pre	102	4.92 (±1.51)	1.00	7.00	−2.86	0.005
Relevant content post	102	5.36 (±1.43)	2.00	7.00		
Annotations pre	102	5.31 (±1.73)	1.00	7.00	−1.53	0.128
Annotations post	102	5.56 (±1.70)	1.00	7.00		

M: mean; SD: standard deviation; N: total sample; Min: minimum; Max: maximum; t: paired samples *t*-test.

## Data Availability

The data are available by contacting the corresponding authors.

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
