# Peer review of "Influence of Gamification and Cooperative Work in Peer, Mixed and Interdisciplinary Teams on Emotional Intelligence, Learning Strategies and Life Goals That Motivate University Students to Study"

_ijerph, 2022, doi:10.3390/ijerph20010547_

Round 1

Reviewer 1 Report

The subject matter of this study is of great relevance and interest scientist. The bibliography used is adequate and up-to-date, not
however, some specific reference to strategies of learning should be included to complete the review of the theoretical framework and the
discussion, in which it is observed how motivation and social skills are included in support techniques.
Inside the classification of study techniques, these are the least studied, which is a strong point of the present study. I think it's
important to highlight this.
The instruments used are adequate to the objectives set being valid and reliable. Table 1 of results should include the T-score to complete the data presented. Just like if you have specific data on the subvariables of each questionnaire (reflected in the instruments section-from line 122 to 126) It is convenient to enter them in the table to provide more information in the analysis of the data obtained.

In the references (for example in the 9, 10, 17, 22, 30 and 34) leave only the first surname of the authors - APA regulations.

Author Response

We thank the reviewer for each and every one of the indications considered in the revision of the manuscript. We have tried to provide a detailed response to each and every one of the indications requested. These suggestions have been reflected in the presentation of the revised manuscript.

Point 1: The subject matter of this study is of great relevance and interest scientist. The bibliography used is adequate and up-to-date, not however, some specific reference to strategies of learning should be included to complete the review of the theoretical framework and the discussion, in which it is observed how motivation and social skills are included in support techniques.

Response 1: We are grateful for the reviewer's suggestions. We have expanded the introduction section, highlighting the importance of learning strategies and their positive influence on the development of social skills (lines 62-66). In addition, reference has been made to the relationship of learning strategies to students' more adaptive motivational patterns (lines 98-100). Following the reviewer's indications, the "discussion" section has been expanded, highlighting recent studies that coincide with our results by pointing out the relationship of motivation to learning strategies (lines 400-404).

Point 2: Inside the classification of study techniques, these are the least studied, which is a strong point of the present study. I think it's important to highlight this.

Response 2: We agree with the reviewer and appreciate his comment. We have expanded the discussion and highlighted the importance of the study technique (lines 420-422).

Point 3: The instruments used are adequate to the objectives set being valid and reliable. Table 1 of results should include the T-score to complete the data presented. Just like if you have specific data on the subvariables of each questionnaire (reflected in the instruments section-from line 122 to 126) It is convenient to enter them in the table to provide more information in the analysis of the data obtained.

Response 3: We are grateful for the reviewer's comments. We have added T-scores to complete the data presented in Table 1 (lines 304-310). We agree with the reviewer's excellent comments on the possibility of including specific data results for the sub-variables of each questionnaire; we have added Table 2, performing the corresponding analysis and presenting the statistical data for the specific differences observed (lines 323-345).

Point 4: In the references (for example in the 9, 10, 17, 22, 30 and 34) leave only the first surname of the authors - APA regulations.

Response 4: We appreciate the reviewer's insight. We fully agree with his comments. We have checked all bibliographic references to ensure that all references meet the stated criteria (lines 479-600).

Point 5: English language and style are fine/minor spell check required

Response 5: To ensure the maximum idiomatic correctness of the text, the manuscript has been revised by MDPI's Language Editing services, thus guaranteeing the correct adaptation of the language.

Reviewer 2 Report

Although this manuscript is quite interesting for the readers, it has some points that need improvements. Here are my suggestions:

- There is no need to present numbers in the abstract section.

- In the 2.3. Procedure section please add a picture of the board game.

- You do not need to 2.4. Study Outcomes section. This section should be moved under the instruments section. Also, there is a need for explaining the sub-factors of each instrument. In the findings section, the authors analyzed the sub-factor means but there is no information about them in the instruments section.

-The discussions section should be developed and the authors should discuss their findings by referencing the current gamification literature. There is a very limited discussion of the findings.

-The conclusions section should be expanded. That shouldn't be your only conclusion from this research.

Author Response

We thank the reviewer for each and every one of the indications considered in the revision of the manuscript. We have tried to provide a detailed response to each and every one of the indications requested. These suggestions have been reflected in the presentation of the revised manuscript.

Point 1: There is no need to present numbers in the abstract section.

Response 1: We agree with the reviewer's comment, we have deleted the numbers in the abstract section (lines 27-29).

Point 2: In the 2.3. Procedure section please add a picture of the board game.

Response 2: Thank you for your comment. We have added the digital board in its original format in Figure 1 (line 220).

Point 3: You do not need to 2.4. Study Outcomes section. This section should be moved under the instruments section. Also, there is a need for explaining the sub-factors of each instrument. In the findings section, the authors analyzed the sub-factor means but there is no information about them in the instruments section.

Response 3: We appreciate the reviewer's suggestion. We have deleted the section "2.4. Study Outcomes" (lines 250-258) and moved it to the section "2.2. We agree with the reviewer and have explained each of the sub-factors in the “instruments” section (lines 158-182).

Point 4: The discussions section should be developed and the authors should discuss their findings by referencing the current gamification literature. There is a very limited discussion of the findings.

Response 4: We are grateful for the reviewer's comments. We have expanded and confronted our results in the discussion section, adding new references (lines 371-375; 380-383; 387-391; 400-404.

Point 5: The conclusions section should be expanded. That shouldn't be your only conclusion from this research.

Response 5: We are grateful for the reviewer's comments in this section. After expanding the results and enlarging the "discussion" section, we have expanded the "conclusions" section (lines 439-457) based on the data from the sub-variables of the instruments used, emphasizing both the academic and employment benefits derived.

Point 6: English language and style are fine/minor spell check required

Response 6: To ensure the maximum idiomatic correctness of the text, the manuscript has been revised by MDPI's Language Editing services, thus guaranteeing the correct adaptation of the language.

Round 2

Reviewer 2 Report

I would like to congratulate the authors for a job well done. Overall this revision has a better balance. It is ready to be accepted.